# Early Development of Direct Embryos in the Cultured Anthers of *Manihot esculenta* Crantz

**DOI:** 10.3390/plants9101315

**Published:** 2020-10-06

**Authors:** Lakmali Dissanayake, Prasanthi Perera, Thilak Attanayaka, Erwin Heberle, Manosha Jayawardhana

**Affiliations:** 1Faculty of Agriculture and Plantation Management, Wayamba University of Sri Lanka, Makandura, Gonawila, 60170 (NWP), Sri Lanka; dmlsdlakmali@yahoo.com (L.D.); dpstg@yahoo.com (T.A.); manosha.lakmini@gmail.com (M.J.); 2Institute of Microbiology and Genetics, Max-Perutz-Laboratories, University of Vienna, Dr. Bohrgasse 9, A-1030 Vienna, Austria; erwin@heberle.at

**Keywords:** anther culture, cassava, proembryos, scanning electron microscope (SEM), simple sequence repeats (SSR)

## Abstract

Cassava is one of the most important sources of energy. To meet the growing demand, genetic improvement is of utmost importance. Its cross-pollinating nature limits the opportunity of exploitation of hybrid vigor and demands the development of homozygous lines through doubled-haploid technologies. The problems in callus-mediated embryogenesis, such as longer processing time and genetically unstable nature, can be overcome by direct embryogenesis. Conditions to produce embryos directly from microspores in cultured anthers were optimized. The optimum stress pretreatment condition was 40 °C for 6 h after culturing the anthers into the induction medium. For proembryo formation, 2% sucrose and 5 mg/l 2,4-dichlorophenoxyacetic acid (2,4-D) or 1 mg/l 1-naphthaleneacetic acid were optimum. Globular embryos were formed by subculturing proembryos into the medium with 0.5 mg/l 2,4-D and 5 mg/l 6-benzylaminopurine after two weeks of culturing. Light microscopy of cultured anthers demonstrated the formation of multicellular structures and their further development into proembryos. Microscopic studies showed proembryos emerging through the damaged anther wall. Monoallelic banding in simple sequence repeat (SSR) analysis indicated homozygous or haploid states in some of the originated embryos. The conditions optimized in this study were effective in the early development of direct embryos after two weeks of culture initiation. This is the first report of the formation of direct embryos in cultured anthers of cassava.

## 1. Introduction

The biological nature of cassava, including drought tolerance, ability to grow in diverse climatic conditions and in depleted soils, its fast recovery from damage caused by pests and diseases, etc. [1,2], makes it a crop with increasing demand, particularly under the global issues of food security and climate change [3]. Starch produced from cassava is amenable for use in various applications, both dietary and industrial [4]. The latter, including pharmaceutical uses, the production of bioethanol and of modified starches for a variety of commercial purposes, caused cassava to transform into the status of a cash crop. In order to fulfill the emerging global demand, however, improvements have to be made in the genetic aspects. 

The genetic improvement of cassava is hampered by the heterozygous nature created by its crossbreed behavior, while the availability of homozygous lines will accelerate the production of F_1_-hybrids with improved traits [5]. It is of the utmost importance to develop efficient protocols for producing pure homozygous lines, where conventional methods will take many years to reach a nearly homozygous state, i.e., nine years in cassava [6]. Doubled-haploid technology is a nonconventional fast approach for this purpose, and many laboratories and breeding companies routinely use it to improve a wide range of crops [7]. 

One doubled-haploid technology is anther or microspore culture where immature haploid male gametophytes are triggered into a sporophytic mode of development to produce haploid or doubled-haploid embryos. Anther culture is cost-efficient, with minimum technological requirements compared to the more advanced microspore culture technique [8]. Microspore embryogenesis has been observed in cultured anthers of various crops, including monocots [9,10,11] and dicots [12,13]. In cassava, both anther and microspore cultures [14,15] have been reported; however, complete success in generating homozygous plants has not yet been achieved. 

Haploid plants can form via two pathways, i.e., indirect embryogenesis through a callus phase and the direct formation of embryos in anther cultures. Calli formation has been reported in a number of crops, such as apricot [16] and peach [17], including cassava [14]. However, direct embryo formation is preferred over callus-mediated regeneration, because calli are generally genetically fragile, i.e., subjected to genetic variations, potentially resulting in a polysomic status of callus-derived plants [18,19]. Furthermore, it has the potential to introduce unwanted genetic changes [20,21] in the genotype of the original gametophytes. Furthermore, since the plants are directly from embryos, several of the procedural steps required in callus-mediated regeneration can be omitted, making the process simpler, shorter and more cost-effective. While optimized conditions for callus formation in cultured anthers of cassava have been reported [14], the conditions for direct embryogenesis have not been reported. 

Thus, the present study was conducted with the overall objective of developing an efficient protocol to induce direct embryogenesis in the cultured anthers with the aim of producing homozygous lines in cassava. Emphasis was given to optimize the critical factors, including heat pretreatment and culture medium components for inducing embryogenesis in the cultured anthers. The study revealed that direct embryogenesis could be induced by applying heat shock to the cultured anthers supplemented with auxins. In addition, the light and scanning electron microscopic and molecular analysis provided the evidence for the induction of direct embryogenesis in the cultured anthers. 

## 2. Results

### 2.1. Optimized Culture Conditions for Inducing Direct Embryogenesis

#### 2.1.1. Effect of Anther Pretreatment

A significant effect was observed among the tested pretreatment conditions (X^2^-15.82; *p* < 0.05; Figure 1a). With a pretreatment of 37 °C, no significant difference was observed among the tested durations; however, the response after 24 h was significantly higher than in the control (X^2^-5.63; *p* < 0.05). The time embryos needed to emerge from the anthers was about three weeks. Anthers incubated at 40 °C at different durations, in contrast, showed a significant time-dependent effect on proembryo formation. Treatments with 40 °C for 6 h and 14 h gave higher anther responses over the non-pretreated anthers (each 5.0%; X^2^-5.63; *p* < 0.05), whereas a lower frequency was observed at longer durations. Importantly, anthers incubated at 40 °C gave rise to proembryos within two weeks of culturing. Thus, this treatment was selected for further testing.

#### 2.1.2. Effect of Anther Pretreatment and the 2,4-Dichlorophenoxyacetic Acid (2,4-D) Concentration in the Medium

The results revealed that the medium without a 2,4-D, interestingly, heat treatment had no effect. Thus, 2,4-D was absolutely necessary. Anther responses among different time durations in the concentrations of 5 (X^2^-14.09; *p* < 0.05), 12 (X^2^-45.69; *p* < 0.0001) and 48 mg/l (X^2^-19.34; *p* < 0.001) showed significant differences (Figure 1b). Another observation was that in the medium containing 5 mg/l 2,4-D, non-pretreated anthers gave the best anther response; however, the time emergence of proembryos was longer than the pretreated ones (data not shown). Therefore, a treatment of 40 °C for 6 h was essential for the early induction of embryogenesis and was selected as the optimum pretreatment condition for further experimentation. 

#### 2.1.3. Effect of 2,4-D and Sucrose Concentration in the Medium

The amount of sucrose in the culture medium had a significant effect on the formation of proembryos at both 5 (X^2^-80.76; *p* < 0.0001) and 12 (X^2^-47.67; *p* < 0.0001)-mg/l 2,4-D (Figure 2a). Medium supplemented with 2% sucrose resulted in the highest anther responses: 27.3% and 13.7% in the media supplemented with 5 and 12-mg/l 2,4-D, respectively, and sucrose concentrations higher than 2% reduced anther response.

#### 2.1.4. Effect of the Other Types and Concentrations of Growth Regulators in the Medium

Significant effects were observed with both hormones 2,4-D (X^2^-13.68; *p* < 0.05) and 1-naphthaleneacetic acid (NAA) (X^2^-14.7; *p* < 0.05; Figure 2b). With 5 mg/l being optimal, 2,4-D had the effect expected from the previous experiments, while NAA at the same concentration was inhibitory, the optimal being the lowest concentration used, i.e., 1 mg/l, as compared to the control without NAA (X^2^-6.62; *p* < 0.05). Overall, however, 2,4-D was more effective than NAA. The experiment on cytokinins revealed that N6-2-isopentyladenine (2iP) was inhibitory on anther response (X^2^-9.65; *p* < 0.05; Figure 2c), while kinetin had no effect. Therefore, further experiments were not performed with these two cytokinins. Within the tested range of concentrations, i.e., 5-15 mg/l, an increasing trend was observed only with 6-benzylaminopurine (BAP) (X^2^-7.87; *p* < 0.05).

In the experiment with 2,4-D and NAA, NAA alone at 1 mg/l had an effect similar to 2,4-D at 5 mg/l, which was greater than the control (X^2^-18.08; *p* < 0.0001 and X^2^-10.22; *p* < 0.05; Table 1). When together in the same medium, NAA and 2,4-D had an additive effect, with 2.5 mg/l 2,4-D plus 0.5 mg/l NAA having an effect similar to 5 mg/l 2,4-D, while 2,4-D at the optimal concentration of 5 mg/l plus 1 mg/l NAA strongly decreased the anther response (X^2^-27.5; *p* < 0.0001). The combination of 2,4-D and BAP, however, showed a significant positive effect (X^2^-103.97; *p* < 0.0001; Table 1). In the experiment with 2,4-D and BAP, the performance of cultured anthers in the hormone-free medium or medium supplemented with 5 mg/l 2,4-D was comparable to the previous experiment, and, like NAA in the previous experiment, BAP alone could replace 2,4-D in its effect on the anther response. When together in the same medium, BAP and 2,4-D had an additive effect that was stronger than NAA plus 2,4-D at the same concentrations, with 2.5 mg/l 2,4-D plus 0.5 mg/l BAP effectively tripling the anther response over a medium with 5 mg/l 2,4-D (X^2^ = 53.78; *p* < 0.0001) and doubling it over 1 mg/l BAP (X^2^ = 29.28; *p* < 0.0001). 2,4-D at the optimal concentration of 5 mg/l plus 1 mg/l BAP still led to an increase in anther response when compared to the two hormones individually. 

#### 2.1.5. Testing the Effect of the Early Subculturing of Anthers for Inducing Direct Embryos 

A significant effect was observed among the media (X^2^ = 123.2; *p* < 0.0001; Figure 3). Subculturing the anthers after two weeks onto a medium supplemented with 0.5 mg/l 2,4-D with 5 mg/l BAP gave rise to the highest number of embryos (28%). These embryos were still globular-shaped and of opaque white color. On the pathway with MS medium with 5 mg/l 2,4 D for two weeks, followed by 0.5 mg/l 2,4-D + 2 mg/l BAP solid medium, the anther response was lower compared to the same with 5 mg/l BAP (X^2^ = 6.82; *p* < 0.01). By maintaining the anthers continuously in the same media, the embryos turned brown.

### 2.2. Microscopic Analysis

#### 2.2.1. Morphological Aspects of Anthers in Culture

Stereo microscopic observations showed that fresh anthers had a smooth anther wall surface in greenish color (Figure 4a). After one week of culturing, all anthers turned to brownish. An enlargement of anthers was observed in some anthers, while the others remained unchanged (Figure 4b). With time, the multicellular structures at the early stage of embryogenesis were visible through the translucent, probably dead, anther wall (Figure 4c). Such proembryos could be observed most of the time after two weeks of culture initiation but, also, up to eight weeks later, indicating that the potential for embryogenesis exists for a considerable period of time. Later, proembryos were visible outside the damaged anther wall (Figure 4d). They were round and had a fragile translucent appearance and a smooth surface. 

By transferring anthers with proembryos of variable sizes onto a solid medium with a low auxin:cytokinin ratio (1:10), the globular embryos converted into a white, opaque appearance (Figure 4e). Most of the responsive anthers gave rise to multiple embryos (Figure 4f), whereas some gave rise to the secondary embryos as well. 

#### 2.2.2. Scanning Electron Microscopic Observation 

Observations on fresh (Figure 5a) and cultured anthers revealed that embryos popped out from a cracked anther wall. Proembryos just emerging through the anther wall clearly revealed the absence of any physical attachment to the anther wall cells (Figure 5b,c). Proembryos and embryos were characterized by globular shapes with a smooth outer surface (Figure 5d). 

#### 2.2.3. Histological Analysis 

The cellular changes occurring in microspores during the transition from gametophytic to sporophytic development after culturing and the further development of multicellular structures were studied. The majority of the microspores in the cultured anthers were dead after two weeks of culturing. Among the remaining alive ones, enlarged microspores (Figure 6a) were observed. On these multicellular structures, the exine was still visible (Figure 6b,c). Multicellular structures at the early globular stage were visible in the cavity of anther lobe after two weeks of culturing (Figure 6d), which were morphologically also visible (Figure 6c). With further growth, these structures appeared to break through the anther wall (Figure 6d). The presence of a protoderm around the embryos indicated the characteristic feature of an organized structure (Figure 6f). Among the analyzed structures, heart-shaped embryos were also observed (Figure 6f). Secondary embryogenesis was also observed among certain anther-derived proembryos. In some embryos, differentiation into the meristematic region (Figure 6h) was observed. Isolated multicellular structures that grew within the anther lobe cavities gave the evidence for their microspore origin.

### 2.3. Simple Sequence Repeat (SSR) Marker Analysis 

Among the screened 18 SSRs, three (SSRY 20, SSRY 94 and SSRY 107) showed a discernible heterozygous pattern with two alleles in the donor plants (Figure 7a). As shown in Figure 7b, SSRY 20 showed a heterozygous locus having two different alleles pertaining to a 200-bp locus, SSRY 94 at a 300-bp locus and SSRY at a 150-bp locus in the parental or donor plants. There was no difference in the SSR patterns observed among four different donor plants of the same variety. Therefore, the leaf DNA sample of one parental plant was used for the analysis. 

Samples with monoallelic bands were observed in the tested two batches of anther-derived structures (batch 1—Figure 7c–e and batch 2—Figure 7f–h) with the selected heterozygous markers for the donor plant (SSRY 20, SSRY 94 and SSRY 107). Four samples from batch 1 (4, 7, 12 and 16) and two samples from batch two (27 and 31) showed a single allele of the respective heterozygous loci for all three primers. Altogether, 18.8% proembryo-derived calli were identified as either haploids or doubled-haploids. 

## 3. Discussion

Considerable efforts have been put into producing cassava doubled haploids by anther and microspore cultures, yet a number of obstacles remain. The pretreatment condition and the culture medium composition are two major factors affecting androgenesis induction [22,23]. Success of the doubled-haploid protocol depends on the efficiency of triggering androgenesis by optimizing the critical factors. 

In many crops, the pretreatment of anthers has shown to be critical for inducing anther response by blocking gametophytic development and triggering pollen embryogenesis [24]. More often, heat shock has been used [25]. In cassava, heat pretreatment at 38 °C was favorable over cold, because cold-treated anthers had a potential to simultaneously give rise to calli from both gametophytic and sporophytic cells [14]. In the present study, higher temperatures of up to 40 °C were tested for the efficient induction of direct embryogenesis. Furthermore, the duration of exposure to each temperature level was also relevant. A heat shock of 40 °C for 6 to 14 h turned out to be optimal. Even though anthers pretreated at 37 °C gave rise to a higher anther frequency, the time embryos needed to emerge from the anther was longer, i.e., three weeks instead of two weeks or less at 40 °C. 

The optimization of culture conditions that induce direct embryogenesis in the cultured anthers is the second step to be optimized. A previous work showed that calli formed in the cultured anthers of cassava genotype SM 1219–9 on a medium supplemented with 8-mg/l 2,4-D [14]. In these experiments, the time needed for androgenesis induction was 8-16 weeks. The present study revealed that direct embryogenesis occurred much more rapidly, i.e., already after two weeks of culturing. Thus, it was possible to save 6–14 weeks for the first step of the protocol. Furthermore, this may be favorable for the genetic stability of anther-derived structures. Whether the shorter time is an effect of the treatment or due to the use of a different genotype remains to be seen. Since the indirect method is not favorable due to several reasons, such as resulting in a mixture of ploidy levels due to the heterogenous nature of the starting materials [26], vulnerability to genetic instability [18,19] and high resource utilization for the maintenance of the cultures for a longer period in the plant regeneration process, the direct embryogenesis from microspores induced in the present study had a higher value for efficient protocol. Anther enlargement was the first significant response observed in cultured anthers that later produced embryos. Anther enlargement prior to embryo formation was observed in some other crop species, including buckwheat [27] and cucumber [28]. 

Culture medium composition, especially growth regulators, energy source, amino acids and activated charcoal, play a critical role in successful anther culture [13,29]. In the present study, the sugar concentration and growth regulators were optimized. Sugar in the tissue culture medium acts as the source of carbon and energy. Sucrose is the most commonly used carbon source for in vitro culture media, and the concentration in the medium affects the induction of androgenesis [28]. It also acts as an osmotic agent [22] to provide a stress condition for inducing androgenesis. In previous reports, 9–10% sucrose was favorable for androgenesis in cassava anther cultures [14,15]. However, the Sri Lankan cassava genotype negatively responded to the increased sucrose concentration. The lower sucrose concentration may be the reason for the shorter time required for emergence of the embryos. 

Auxins and cytokinins are critical molecules for plant growth and development by controlling cell division and morphogenesis [16,30]. The most commonly used auxin in anther cultures is 2,4-D, and the optimum concentration required is species-dependent [31,32]. In other crops, the early exposure of proembryos into higher concentrations of cytokinin has been reported to be effective in converting the embryos [9]. In our experiments, 5 mg/l 2,4-D turned out to be optimal to produce proembryos. Higher concentrations suppressed anther response. Among the other growth regulators, NAA and BAP had, within limits, a positive effect and an additive effect with 2,4-D on anther responses. The combinations of 2,4-D at 5 mg/l and NAA or BAP at a concentration of 1 mg/l, respectively, gave significantly lower proembryo formations over the concentrations of 2.5 mg/l and 0.5 mg/l. Therefore, the choice of concentration should be done carefully, since the combining effect can have a negative impact on androgenesis induction. However, even though a large number of proembryos were initiated, they did not convert into plants under same the culture conditions. The reason could be a carryover effect of the auxins in the induction medium.

Exposing of the proembryos at early stages to cytokinin was effective in producing embryos with globular shapes in ivory/opaque colors. After subculturing anthers taken from two-week-old anther cultures at 5 mg/l 2,4-D on a medium with reduced 2,4-D (0.5 mg/l) and increased BAP concentration (2 or 5 mg/l), the shoot and root initiation were observed in histological sections revealing the conversion of globular embryos into heart-shaped embryos occasionally occurred. However, plant regeneration was not achieved. Yet, the reproducible protocol developed in this study can be the basis for further improvements of the plant regeneration protocol. 

Observations by stereo, light and scanning electron microscopy showed that, in responsive anthers, the anther wall broke, and embryos came out from the inside of the anther lobe. This was a clear indication that these embryos were of microspore origin. To distinguish between somatic and haploid embryos, a SSR analysis was used. This technique is the most informative molecular marker system among the various DNA marker methods. SSR markers are codominant, allowing the identification of both alleles of a heterozygous locus. In the present setting, by studying the SSR loci, the genotype of an anther donor was determined, and somatic (heterozygous) and haploid/homozygous embryos were distinguished. SSR marker analysis has been proven to be a suitable tool for identifying the origin of calli in cassava microspore cultures [14]. 

Among the 32 proembryo-derived calli analyzed through SSR markers, the majority was polymorphic with heterozygous banding, while only six were monoallelic (19%). The latter calli may have been either haploids or doubled haploids due to spontaneous diploidization. A polymorphic banding pattern can be due to several reasons. First, the analyzed sample may have originated from the connective tissue or anther wall. However, attempts were made to remove the somatic proembryos at a very early stage. A second possibility is that two multicellular structures derived from different microspores mixed when the callus proliferation was induced in the proembryo. The histological micrographs also provided evidence for this hypothesis. The proembryos were indeed seen close together in the anther lobe. Thirdly, embryos with a heterozygous allele status can also originate from unreduced microspores that have been reported to occur with comparatively high frequency in cassava [33]. Similar work observation was done in the genotype *Kirikawadi* used for the present study. 

## 4. Materials and Methods 

### 4.1. Bud Collection, Sterilization and Culture Initiation

Cassava genotype *Kirikawadi*, which produces flowers throughout the year, was used as anther donors. *Kirikawadi* is a commercial variety recommended by the Department of Agriculture, Sri Lanka in 1994 as a selection of local germplasm. Inflorescences were picked before 09.00 a.m. and sealed in zip bags and immediately brought to the laboratory. Immature male buds ranging from 1.3 to 2.0 mm in diameter were selected, since they contained the tetrads or early uninucleate microspores that were reported to be responsive for cassava androgenesis [14,15]. 

For surface sterilization, the selected male buds were washed with soap water and kept under the running tap water for about 30 min. Cleaned buds were subjected to further disinfection under the laminar flow hood (Flow FAST V, Serrara, Italy) using 70% ethanol for 1 min, followed by 10% Clorox (*v/v*) with 60 μl/l Tween 20 for another 10 min. Disinfected buds were thoroughly washed three times for 2 min with sterilized water and kept in a sterile Petri dish containing a wetted white paper to be used immediately for culturing. A stereo microscope (Euromax, Arnhem, The Netherlands) was used to excise the anthers and cultured into the media as described below.

### 4.2. Basal Culture Medium

Murashige and Skoog medium (MS) [34] modified by supplementing 0.225 mg/l CuSO4, 64.2 mg/l NaFe EDTA and 100-mg/l myoinositol was used as basal medium. Tissue culture-tested chemicals and growth hormones (Sigma-Aldrich, Missouri, USA) were used for preparing the media. Growth regulators were added to the basal medium according to the experimental requirements. Sucrose (2%; w/v) was used as the energy source unless mentioned otherwise. The pH of the media was set to 5.8 with 1-N KOH using a pH meter (OHAUS, Columbia, USA). The culture media were sterilized by autoclaving at 121 °C (Hirayama, Tokyo, Japan), and culture plates were prepared. The excised anthers were carefully placed on the surface of the liquid culture medium in Petri dishes (90 × 15 mm) containing 25 ml of liquid culture medium.

### 4.3. Optimization of the Culture Conditions for Inducing Direct Embryogenesis

#### 4.3.1. Testing the Anther Pretreatment

The anthers were cultured into the medium containing 5 mg/l 2,4-dichlorophenoxyacetic acid (2,4-D). The cultures were incubated at two temperature levels (37 °C and 40 °C) using two incubators (Memmert IN260, Schwabach, Germany and Memmert ICP55, Schwabach, Germany) for four time periods (6, 14, 24 and 48 h) and tested against the control of non-pretreated anthers at 28 ± 2 °C. 

#### 4.3.2. Testing the Anther Pretreatment and the 2,4-D Concentration in the Medium

It was tested whether the 2,4-D concentration and the pretreatment durations were optimal. Anthers were cultured into the media supplemented with different 2,4-D concentrations (0, 5, 12, 24 and 48 mg/l) and then incubated at 40 °C for different time durations (0, 6, 14, 24 and 48 h).

#### 4.3.3. Testing the 2,4-D and Sucrose Concentration in the Medium

For further optimizing the 2,4-D level, 5 and 12 mg/l were tested with four levels of sucrose (2%, 9%, 12% and 20%) in MS media. Cultured anthers were pretreated at 40 °C for 6 h and then maintained in the dark at 28 ± 2 °C.

#### 4.3.4. Testing the Effect of the Other Types and Concentrations of Growth Regulators in the Medium

As 2,4-D had a strong effect on the anther response, other auxins were tested for enhancing the anther response. To test the relative strength of 1-naphthaleneacetic acid (NAA) on the anther response, identical concentrations of 2,4-D and NAA (0, 1, 5 and 10 mg/l) were tested. Three cytokinins: N6-2-isopentyladenine (2iP), 6-benzylaminopurine (BAP) and kinetin, with identical concentrations (0, 5, 10 and 15 mg/l) were also tested for inducing direct embryogenesis. Further, the effect of growth regulators was further tested by combining 2,4-D with NAA and BAP. Hormone-free condition and 5 mg/l 2,4-D were included into both experiments. With NAA, 1 mg/l NAA alone, 2.5 mg/l 2,4-D with 0.5 mg/l NAA and 5 mg/l 2,4-D with 1 mg/l NAA were tested. Same concentrations and combinations were tested with BAP. Cultured anthers were pretreated at 40 °C for 6 h and then maintained in the dark at 28 ± 2 °C.

#### 4.3.5. Testing the Effect of Early Subculturing of Anthers for Inducing Regeneration in the Proembryos 

On the above experiments, the proembryos formed did not progress further to form embryos with cotyledons and shoot and root meristems. Thus, in this experiment, the cultures were either maintained continuously in the same induction medium or changed the medium by subculturing into a medium with different levels of growth regulators. The tested six protocols were: no growth regulators in continuous culture (control), MS medium with 5 mg/l 2,4-D in continuous culture (optimized in the above experiments), MS medium with 5 mg/l 2,4 D for two weeks, followed by 0.5 mg/l 2,4-D + 2 mg/l BAP solid medium, MS medium with 5 mg/l 2,4 D/for two weeks and 0.5 mg/l 2,4-D + 5 mg/l BAP solid medium, MS medium with 0.5 mg/l 2,4-D + 2 mg/l BAP in continuous liquid culture and MS medium with 0.5 mg/l 2,4-D + 5-mg/l BAP in continuous liquid culture. Media were solidified with 2.5% (w/v) phytagel as per treatment. Cultures were maintained at 28 ± 2 °C in a 16-h light/8-h dark photoperiod, with the light intensity of 3000 lux.

All experiments were repeated three times. A total of 360 anthers cultured in nine Petri plates (90 × 15 mm; each containing 40 anthers) were used for each treatment of all experiments mentioned above. The cultures were observed at weekly intervals. Responsive anthers were detected under a stereo microscope. The anthers with proembryos emerging by breaking through the anther wall were recorded as responsive ones. Proembryos were defined as globular structures with a smooth surface. Anther response was defined as the proembryo-bearing anthers per 100 cultured anthers. 

### 4.4. Data Analysis

Chi-square or maximum likelihood analysis of variance was conducted using the Proc CatMod procedures of PC-SAS [35] to analyze the count data generated from the experiments. Treatment means were compared using SE, 95% confidence intervals or orthogonal contrast coefficients [36]. 

### 4.5. Microscopic Analysis

Ten anthers from each fresh and cultured anther at 2, 4 and 6 weeks of culturing were sampled for microscopic observation. Samples were fixed in FAA (50% ethanol + 10% formaldehyde + glacial acetic acid, 18:1:1) for 72 h and then dehydrated through a graded ethanol series (30, 50 and 70%, *v/v*) for 2 h by changing the solution at each hour. Samples were stored in 70% ethanol until use. Fixed samples were used for scanning electron and light microscopic analyses.

#### 4.5.1. Scanning Electron Microscopic Analysis 

The fixed specimens were mounted onto aluminum cylinders and observed using a scanning electron microscope (JEOL JSM820, Tokyo, Japan). Images were captured using a Nikon D3000 in a Taylor adaptation system (M.E. Taylor Engineering Inc., MD, USA). 

#### 4.5.2. Light Microscopic Analyses

Samples stored in 70% ethanol were further dehydrated through a graded ethanol series (80, 90, 95 and 100%, *v/v*) for 2 h by changing the solution at each hour and 100% butanol for 48 h. They were preinfiltrated with butanol:resin (1:1) solution for 24 h and embedded in resin according to the manufacturer’s guidelines. Prepared histoforms were fixed onto histoblocks. Sections at 4 to 5-μm thickness were obtained using a rotary microtome (Microteck, Ettlingen, Germany) and loaded onto glass slides maintained at about 60 °C. Prepared slides were stained with protein-specific 5% Naphthol Blue Black (Sigma-Aldrich, St. Louis, USA) for 5 min at 60 °C, washed three times with double-distilled water and air-dried. After placing coverslips, all sections were observed under a light microscope (Optika, Ponteranica, Italy) and photographed with an attached camera (Optika, Ponteranica, Italy).

### 4.6. Simple Sequence Repeat (SSR) Marker Analysis

In order to increase the cell mass to make them sufficient for DNA analysis, callogenesis was induced in anther-derived proembryos by subculturing them on the same medium containing 12 mg/l 2,4-D solidified with 2.5% phytagel. Those proembryo-derived callus lines obtained at different time periods named with different batch numbers were used for the following analysis.

Samples were analyzed to check their zygosity status using SSR markers in two batches: batch 1 and batch 2, each containing 16 callus lines representing 16 anther-derived proembryos. Confirmation of the haploid allele status of the proembryos was done by comparing the alleles of the heterozygous SSR loci in the donor plant. Eighteen cassava SSR markers were screened for the donor genotype to select the markers with heterozygous allele status.

Genomic DNA was extracted from the tender leaf of four different cassava donor plants used for collection of anthers and from the proembryo-derived calli. About 20–50 mg of sample was homogenized with 1-ml DNA extraction buffer (1-M Tris-HCl, 4-M NaCl and 0.5-M EDTA) in a centrifuge vial. The supernatant (~500 µl) was taken after centrifugation at 9800 rpm for 5 min and incubated at 65 °C for 15 min. Then, 500 µl of chloroform was added, vortexed and centrifuged at 13,000 rpm for 10 min at 4 °C. About 200 µl was taken from supernatant and mixed with 200 µl of ice-cold isopropanol. After keeping the mixture for 30 min at −20 °C, it was centrifuged at 13,000 rpm for 15 min in 4 °C. Precipitated DNA pellet was washed with 1000 µl of 70% ice-cold ethanol by centrifugation. It was dried for 2 h at room temperature and dissolved in 50 µl of double-distilled water. DNA was stored at −20 °C, and 10 µl of DNA was taken for PCR amplification.

The primer sequences of the cassava SSR markers used in this study are available in CIAT’s Cassava Molecular Diversity Network (http://newapp.ciat.cgiar.org/molcas/). Eighteen primers (Appendix A) were screened with the extracted donor DNA. PCRs were performed as described by Rampling et al. [37]. Each PCR reaction contained 1X PCR buffer, 1.5-mM MgCl_2_, 200-µl dNTPs, 5-µmol each forward and reverse primers and 1-U GoTaq (Promega WI, USA) and 50 ng of DNA. After performing PCR (Appendix A), the products (4 µl) were electrophoresed on 1.5% ultrapure agarose gel prepared using 0.5X TBE (tris borate EDTA) buffer. The gel was run at 100 V/cm for 45–60 min in an agarose gel electrophoresis unit (Enduro^TM^, NJ, USA), USA) using the same buffer. DNA were stained with 0.4 µl of ethidium bromide and observed under a UV transilluminator with the help of a UV-protected gel documentation system (Vilber, Marne-la-Vallée, France).

## 5. Conclusions

The study revealed that direct microspore embryogenesis is possible in cultured anthers. By providing heat pretreatment for the cultured anthers in the medium supplemented with 1.12 mg/l 2,4-D, 1 mg/l NAA and 2.5 mg/l 2,4-D in combination with 0.5 mg/l NAA was effective. However, 2.5 mg/l 2,4-D in combination with 0.5 mg/L BAP was the best medium for inducing microspore embryogenesis where pretreated at 40 °C for 6 h after culturing. However, differentiation into embryos with shoot meristems only occurred by early exposure of the embryos to BAP, whereas the loss of embryogenesis took place when embryos were maintained in the induction medium. The conditions optimized in this study were effective in the early development of direct embryos after two weeks of culture initiation. This is the first report of the formation of direct embryos in cultured anthers of cassava.

## Figures and Tables

**Figure 1 plants-09-01315-f001:**
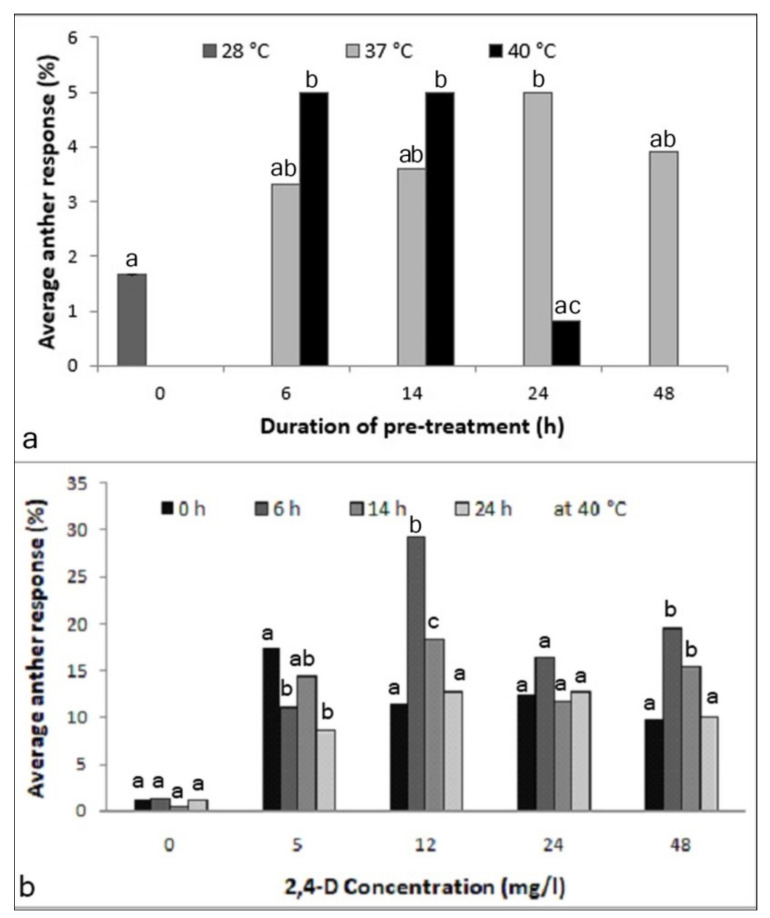
Optimization of anther pretreatment for inducing androgenesis in the cultured anthers of *Manihot esculenta* var. *Kirikawadi*. (**a**) Effect of the incubation temperature and duration (X^2^-15.82; *p* < 0.0451). (**b**) Effect of the time duration at 40 °C in media supplemented with different 2,4-dichlorophenoxyacetic acid (2,4-D) concentrations: 0 (NS), 5 (X^2^-14.09; *p* < 0.0028), 12 (X^2^-45.69; *p* < 0.0001), 24 (NS) and 48 mg/l (X^2^-19.34; *p* < 0.0002). Different letters above the bars at each incubation time indicate significantly different values; n = 360. NS—nonsignificant.

**Figure 2 plants-09-01315-f002:**
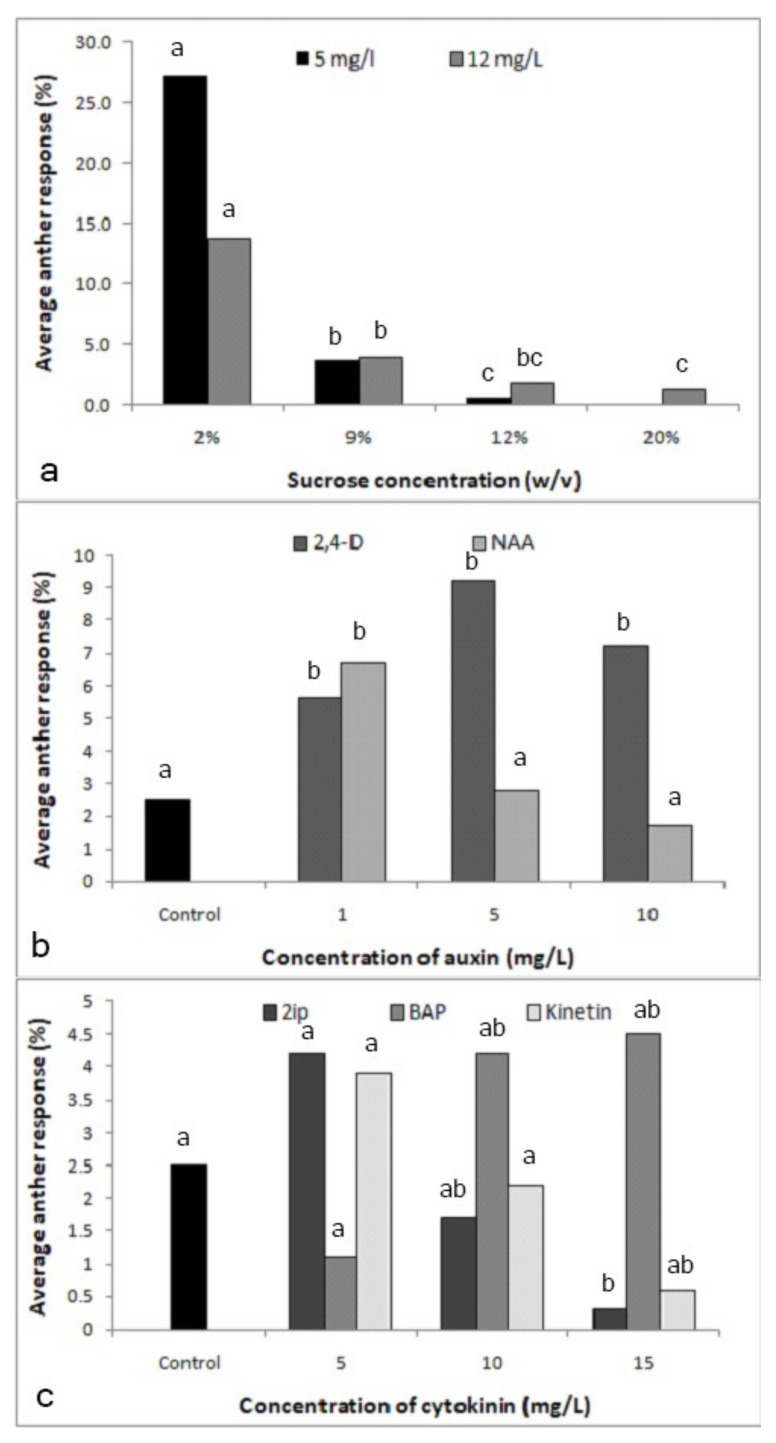
Effect of the culture medium components on the response of anthers cultured in media supplemented with different components. (**a**) Effect of sucrose concentration with 2,4-dichlorophenoxyacetic acid (2,4-D) at 5 mg/l (X^2^-80.76; *p* < 0.0001) and 12 mg/l (X2-47.67; *p* < 0.0001) (Different letters above each bar of each 2,4-D level indicate significantly different values). (**b**) Effect of auxins 2,4-D (X^2^-13.68; *p* < 0.0034) and 1-naphthaleneacetic acid (NAA) (X2-14.7; *p* < 0.0021) (Different letters above each bar of each auxin indicate significantly different values). (**c**) Effect of cytokinins, N6-2-isopentyladenine (2iP) (X^2^-9.65; *p* < 0.0218), 6-benzylaminopurine (BAP) (X^2^-7.87; *p* < 0.0488) and kinetin (X2-7.39; *p* < 0.0606) (Different letters above each bar of each cytokinin indicate significantly different values); n = 360.

**Figure 3 plants-09-01315-f003:**
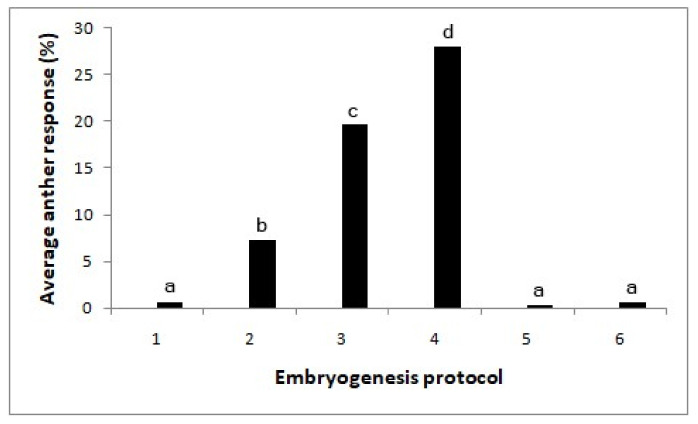
Effect of the subculturing of cultured anthers into media with different hormones on embryo formation (X^2^-123.2; *p* < 0.0001). Different letters above each bar indicate significantly different values. T1—without growth regulators, T2—MS medium with 5 mg/l 2,4-D, T3—MS medium with 5 mg/l 2,4 D/(after two weeks) 0.5 mg/l 2,4-D + 2 mg/l BAP, T4—MS medium with 5 mg/l 2,4 D/(after two weeks) 0.5 mg/l 2,4-D + 5 mg/l BAP, T5—MS medium with 0.5 mg/l 2,4-D + 2 mg/l BAP and T6—MS medium with 0.5 mg/l 2,4-D+ 5 mg/l BAP. N = 360.

**Figure 4 plants-09-01315-f004:**
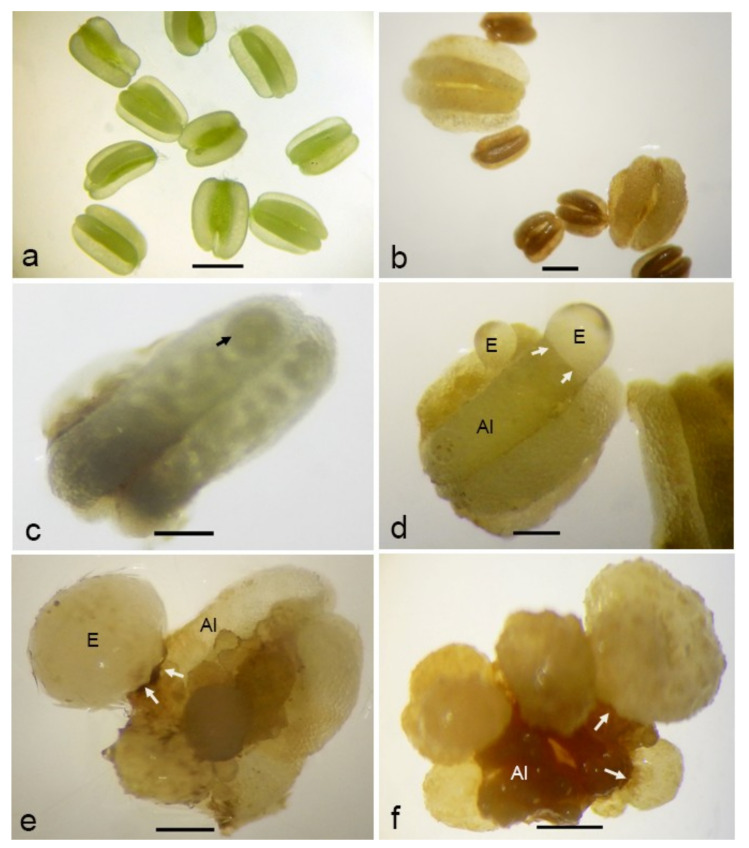
Morphological aspects of anther-derived proembryos of *Manihot esculenta* var. *Kirikawadi*. (**a**) Fresh anthers isolated from the male flower buds. (**b**) Enlarged anthers. Note the nonresponsive ones showing anther browning. (**c**) Visualizing a multicellular structure (arrow) enclosed in the anther lobe. (**d**) Translucent proembryo with the smooth surface emerging through the anther lobe (Al). (**e**) A proembryo converted into ivory color. (**f**) Close view of proembryos emerging through the anther wall. Arrows indicate the remnants of the anther wall (Aw) after emergence of the embryo through the broken anther wall. Bars: (**a**,**b**,**f**) 1 mm, (**c**) 670 µm, (**d**) 500 µm and (**e**) 600 µm.

**Figure 5 plants-09-01315-f005:**
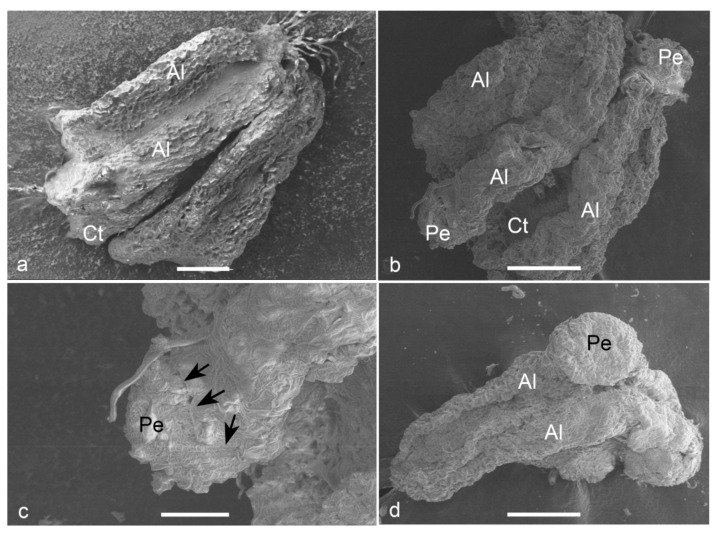
Scanning electron micrographs (SEM) of the anther-derived proembryos of *Manihot esculenta* var. *Kirikawadi*. (**a**) A fresh anther (Al—anther lobe and Ct—connective tissue). (**b**) The proembryos (Pe) emerging through the anther wall. (**c**) Close view of a proembryo emerging from the anther lobe of left side in b. Black arrows indicate the broken anther wall edge. (**d**) Globular embryo with a smooth surface. Bars: (**a**) 100 µm, (**b**,**d**) 200 µm and (**c**) 80 µm.

**Figure 6 plants-09-01315-f006:**
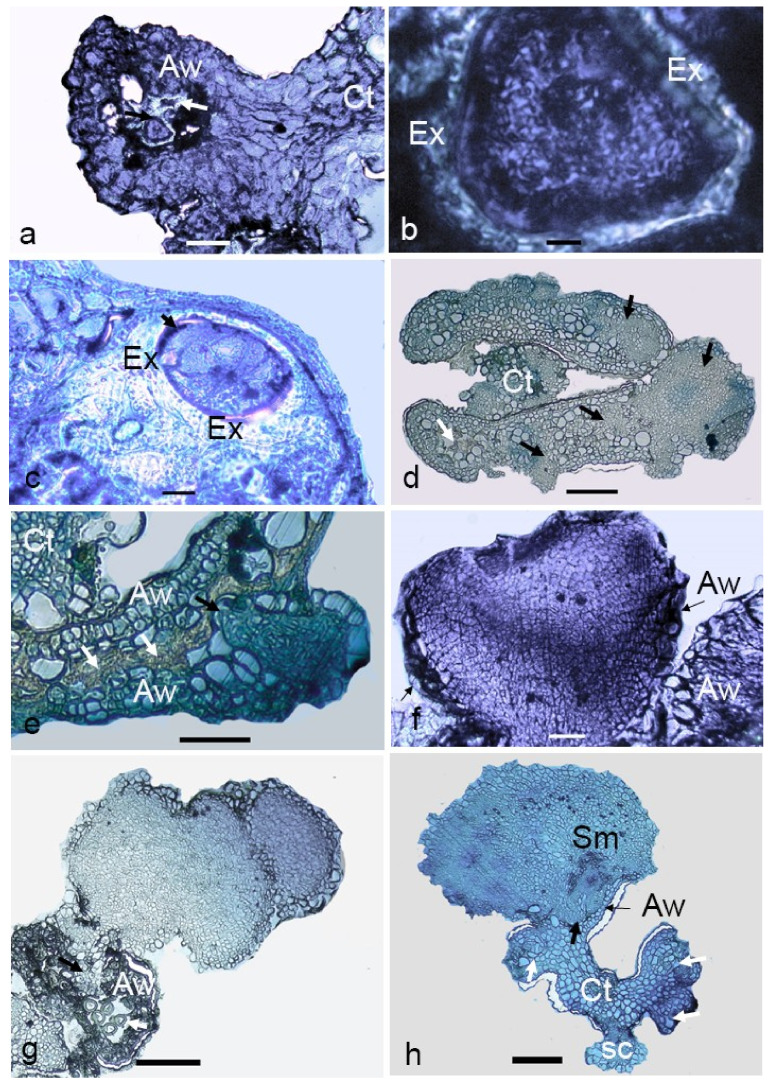
Light micrographs of anther-derived structures of *Manihot esculenta* var. *Kirikawadi*. (**a**) Viable microspores enclosed in the anther wall (Aw) (white arrows indicate nonviable ones) (Ct—connective tissue). (**b**) Close view of viable microspore with thick exine (Ex). (**c**) Microspore where cell division occurred inside the exine. (**d**) Transverse section of an anther containing proembryos (black arrows). (**e**) Proembryo emerging through the anther wall. Black arrows. (**f**) Heart-shaped embryo. (**g**) Secondary embryogenesis occurring on the anther-derived embryo. (**h**) A globular embryo with differentiated root and shoot poles. Note the shoot meristem developing in the haustorial tissue, the anther wall remnants at the emerging point and the somatic structure derived from the connective tissue (Ct). Bars: (**a**,**f**) 100 µm, (**b**) 10 µm, (**c**) 160 µm, (**d**,**e**) 120 µm, (**g**) 60 µm and (**h**) 800 µm.

**Figure 7 plants-09-01315-f007:**
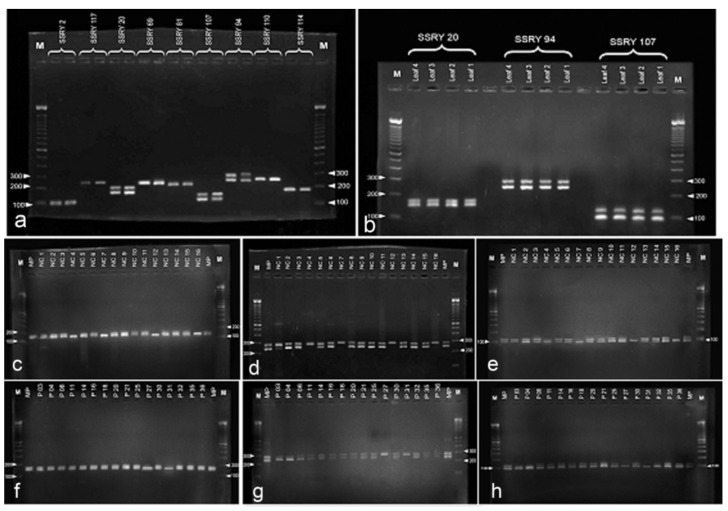
DNA analysis of anther derived structures of *Manihot esculenta* var. *Kirikawadi*. (**a**–**h**) Simple sequence repeat SSR marker analysis. Out of nine SSR primes tested, (**a**) three primers SSR 20, 94 and 107 showed segregating alleles with donor plant DNA. (**b**) Two batches of anther-derived structures, (**c**–**e**) batch 1 and (**f**–**h**) batch 2, were analyzed with the selected three primers consecutively. Note four monoallelic bands in batch 1 and two bands in batch 2.

**Table 1 plants-09-01315-t001:** Effect of the combinations of 2,4-dichlorophenoxyacetic acid (2,4-D) with 1-naphthaleneacetic acid (NAA) and 6-benzylaminopurine (BAP) on anther response of *Manihot esculenta* var. *Kirikawadi*.

Tested GH Combinations (mg/l)	Average Anther Response (%)
2,4-D and NAA	
Control (0 GH)	10^a^
5 mg/l-2,4-D	23^b^
1 mg/l-NAA	28^b^
2.5 mg/l 2,4-D + 0.5 mg/l NAA	31^b^
5 mg/l 2,4-D + 1 mg/l NAA	7^a^
*P*-value	<0.0001
X^2^	45.83
2,4-D and BAP	
0 (0 GH)	10^a^
5 mg/l-2,4-D	22^b^
1 mg/l-BAP	32^bc^
2.5 mg/l 2,4-D + 0.5 mg/l BAP	61^e^
5 mg/l 2,4-D + 1 mg/l BAP	37^c^
*P*-value	<0.0001
X^2^	103.97

GH—growth hormone; averages in a column with same letters are not significantly different at *p* < 0.05; n = 360.

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
