# Peer review of "Early Development of Direct Embryos in the Cultured Anthers of Manihot esculenta Crantz"

_plants, 2020, doi:10.3390/plants9101315_

Round 1

Reviewer 1 Report

Dear Authors,

     I have reviewed this manuscript and I think it is worth to be publish although some points should be improved:

  • First of all, some methodological details should be included for example:

In Materials and Methods chapter there is written that „Cassava genotype Kirikawadi, which produces flowers throughout the year, was used as anther donors.”, but it is not easy to understand what about the origin of analyzed Cassava genotype. Where the plant material comes from?  What about the acessions number? Please explain.

  • In 423 - 425 lines there is written that: „The excised anthers were carefully placed on the surface of the liquid culture medium in Petri dished (90x15 mm) containing 25 ml of liquid culture medium”. But there is no mentioned even about the number of excised anthers placed in one Petri dish and about numbers of Petri dishes treated as one treatment/one repetition. How many Petri dishes were treated as one combination (time period and one level of temperature) during anthers pre-treatment observations?
  • 474 line - For microscopic analysis what do you mean saying „from each stages”. It is not clear 10 anthers from which stages were used?
  • 495 line – please indicate clearly how do you understand the description:” The pro-embryo-derived callus lines”.

- Fig.7 there is probably a typing error in word

c-e) bactch 1 , please rewrite it         

- 424 line – it should be written in Petri dishes instead of „in Petri dished”          

Reviewer 2 Report

1.The present study investigated the problems in direct embryogenesis from anthers of Manihot esculenta. The conditions of effective embryogenesis were optimized for two weeks of culture initiation
2. In the study, the authors establish the influence of plant hormones on direct embryogenesis. This is the first report of the formation of direct embryos in cultured anthers of cassava.
3. I think that the paper is well written and the text is clear and easy to read.
4. The authors well defend statements and support them with evidence. I also think that the results obtained would be of benefit to researchers working in the field of plant biotechnology.
5. Description/statistical significance about a”, “b”,…. “e” should be added in legends of Figures 1 to 3.
These arguments give me a reason to propose a manuscript for acceptance for publication after minor revision.

Reviewer 3 Report

"This is an interesting and well written manuscript concerning the developing of an efficient protocol to induce direct embryogenesis in the cultured anthers with the aim of producing homozygous lines in cassava. This is the first report, as far as I know, of the formation of direct embryos in cultured anthers of cassava in a very short time (two weeks of culture initiation). This is also very important for production of homozygous lines in this species. The text is clear and easy to read and the conclusions consistent with the evidence and arguments presented in the manuscript. Thus, I think that the manuscript could be accepted for publication after minor revision. Below are some minor remarks to the manuscript:

line 462. please add light intensity and, if there is, range in the temperature i.e 28+-2 oC. the same to all the text"
